# Experimental Investigation of the Tribological Behaviors of Carbon Fiber Reinforced Polymer Composites under Boundary Lubrication

**DOI:** 10.3390/polym14183716

**Published:** 2022-09-06

**Authors:** Corina Birleanu, Marius Pustan, Grigore Pop, Mircea Cioaza, Florin Popa, Lucian Lazarescu, Glad Contiu

**Affiliations:** 1MicroNano Systems Laboratory, Technical University of Cluj-Napoca, Blv. Muncii Nr. 103–105, 400641 Cluj-Napoca, Romania; 2Department of Design Engineering and Robotics, Technical University of Cluj-Napoca, 400641 Cluj-Napoca, Romania; 3Department of Materials Science and Engineering, Technical University of Cluj-Napoca, 400641 Cluj-Napoca, Romania; 4Department of Manufacturing Engineering, Technical University from Cluj-Napoca, 400641 Cluj-Napoca, Romania

**Keywords:** composite material, tribology properties, boundary lubrication, stick-slip, friction, wear

## Abstract

Friction and wear experiments were performed on carbon fiber-reinforced polymer (CFRP) composites, and the tribological behavior of these materials under boundary lubrication (based on the 5100 4T 10 W-30 engine oil with TiO_2_ Degussa P25 nanoparticles) was investigated. Experiments were carried out in two directions: one at a different normal load from 6 to 16 N and one at a low sliding speed of 110 mm/min under boundary lubrication conditions. The obtained results reveal the stick-slip effect and the static and dynamic coefficient of friction decreased slightly with increasing normal applied load on the carbon fiber reinforced polymer composite pairs. The second direction highlights through experimental tests on the pin on disc tribometer that the friction coefficient increases with the increase in normal load (20–80 N) and sliding velocity (0.4–2.4 m/s). On the other hand, it is found that the friction coefficient is slightly lower than in the stick-slip phase. During the running-in process, the friction coefficient of the CFRP pair increases steadily as the rubbing time increases, and after a certain rubbing period, it remains constant regardless of the material of the counter face. The obtained results show that for the observed interval, the influence of normal load and sliding velocity have relatively small fraction coefficients and low wear depths. A 3D analysis of the profile demonstrated the texture of wear marks and tracks of these engineering composite materials. Furthermore, the height variations of wear marks and the morphologies of the worn surfaces of specimens under boundary lubrication conditions were analyzed.

## 1. Introduction

Carbon fiber composites are defined as fibers that contain at least 92% by weight carbon, while a fiber that contains at least 99% carbon is usually called a graphite fiber [1]. Carbon fibers generally have excellent tensile properties, low densities, high thermal and chemical stability in the absence of oxidizing agents, good thermal and electrical conductivity, and excellent creep resistance [1].

In recent years, due to its excellent properties, the carbon fiber industry has grown steadily to meet the demand of various categories of industry, such as aerospace, military, construction (non-structural and structural systems), turbine blades, light cylinders, and pressure vessels, drilling elements, medical items, cars, sports, etc. [1]. Specifically, the composites market is expected to reach an estimated value of approximately USD 40 billion by 2024 [2].

For the automotive industry, carbon fiber-reinforced polymer (CFRP) composites offer low weight and superior styling [3,4,5]. Carbon fibers find their applicability in body parts (doors, bonnets, deck covers, front, bumpers, etc.), chassis and suspension systems (e.g., lamellar springs), and drive shafts, where high strength-to-weight ratios are required. One of the main reasons why composite carbon fibers have so much potential in industry is also due to the microstructures of the fibers, as well as their orientation, length, and concentration. Because high damping of carbon fiber-epoxy composite materials can dissipate any vibration of the composite structure that is induced, they are used in the manufacture of high-speed drive shafts [3], machine tool shafts [4], and robot arms [5]. They are usually made in a variety of types of fiber preforms that are impregnated with the two liquid precursors (pitch or resin) before carbonizing a carbon-bearing gas precursor (e.g., methane) to a certain level, with the accepted porosity [6,7,8]. These combinations result in a variety of materials with high specific strength, self-lubricating capacity, low coefficient of thermal expansion, stiffness, and toughness, as well as very good refractory properties [9].

In the last decade, several researchers have been concerned with the friction and wear investigations of different types of materials under different operating conditions. These researchers [1,2,10,11,12,13,14] showed that the friction and wear of self-rubbing polymers and composites depend on several parameters, such as normal load, sliding speed, the roughness of friction surfaces, relative humidity, lubrication, etc. [8,15,16,17,18,19]. Among these parameters, it has been reported that the most influential parameters dictating the tribological properties of materials are normal load and sliding velocity [8]. The coefficient of friction of polymers and composites which slip towards themselves increases or decreases, depending on the range of operating conditions and the sliding pairs that have been chosen. 

For intellectual property reasons and because they are industrial recipes, most of the research on friction and wear of CFRP composites is not published in specialized literature. Relatively high friction coefficients, about 0.34 [20] or in the range 0.4–0.9 [21], are reported for different conditions. Sometimes the wear coefficients indicated in the same works are contradictory: K ≈ 2 × 10^−6^ mm^3^ N^−1^ m^−1^ [22], denoting mild wear, or 1 × 10^−3^ < K < 3 × 10^−2^ mm^3^ N^−1^ m^−1^ [13], corresponding to a highly severe wear regime. Two morphological types of wear are described as follows [20,21,22]: (i) an apparently polished appearance, corresponding to the application of a harmonious layer of wear residues resulting in an almost flat surface because of this phenomenon; (ii) a coarse morphological structure, resulting from uneven wear of the composite matrix and low-adhesion fibers or powder residues (‘dusting’). The first type of residue film has a lubricating effect and reduces the contact pressure, thus reducing both the coefficients of friction and wear [12]. By analogy with conventional graphite-carbon materials, CFRP composites have distinct friction and wear behavior, due to the surrounding environment, i.e., respectively the relative humidity [20,21,22,23,24].

It is known that stick-slip (SS) motion is commonly present at low sliding velocities (under 180 mm/min) when two solid bodies slide over each other [10]. Stick-slip phenomena are all around us, often with both good and bad consequences. This effect is undesirable in many and varied engineering and technological applications due to the jerky motion (stick and slip) which results in vibration and noise, energy loss, wear on slippery surfaces, and can lead to damage to moving components [10].

In the stick stage, the two surfaces in contact do not move and are held in place by static friction. In the sliding stage, there is a finite relative motion—the so-called kinetic friction acts to delay this motion. However, what makes a given system work is that, in general, static friction is greater than kinetic friction. The roughness of the surface as well as the presence of a lubricant (which can even be a film of a softer material) can affect the amplitude and rate of variation of the cycle of the stick-slip phenomenon. These processes take place mostly in the boundary lubrication regime [11].

Tribological contacts between plastic or polymer materials can result in sliding friction behavior that produces noise during operation. The applied load, roughness, stiffness, temperature, and humidity of the working environment are other factors that influence the stick-slip phenomenon. [10,11]. In dry conditions as debated in [25,26,27,28,29], a continually reversing load applied by the moving counter body has a strong influence on friction and wear behavior. Schön [30] measured the μ for composite in contact with composite in reciprocal sliding to model bolted joints of CF/epoxy material, finding a starting value of 0.65 and a peak of 0.74. Klingshirn et al. [31] investigated the influence of the volume content of short carbon fibers inside an epoxy matrix on the fretting wear rate, demonstrating the dramatic decrease in this parameter with growing fiber content.

A lubricant is also generally used to reduce the frictional force during SS movement [2,3,4]. An alternative approach to friction control is the use of nanoparticles that act as a roller and spacer at sliding interfaces [14,20,21,22,23]. For reducing friction and wear, especially in boundary lubrication as well as mixed boundary regimes, nanoparticles are quite effective [7,8,11]. The addition of nanoparticles in a lubricant improves the tribological properties because of ball bearing, polishing, repair, and formation of the protective film [10,14]. Reviewing the specialized literature shows that the delivery and adhesion of particles to the friction surfaces are strongly dependent on the nature of the friction contact and the presence of fluid films on the separation surface [12]. Studies on the tribological properties of nano lubricants at high slip rates and high normal loads show that the anti-friction, anti-wear, and extreme pressure properties are improved in the presence of nano lubricants [10,11,12,13,14,32,33].

Yamamoto et al. [23] reported that under boundary lubrication, the addition of carbon fiber to CFRP made the wear characteristics excellent. Sinmazcelik and Yilmaz studied the effects of thermal aging on the properties of CFRP composite material reinforced with unfilled and randomly oriented short fibers [24]. They evaluated the effects of the microstructural changes, resulting from the thermal aging process, on the tribological, thermal, thermomechanical, physical, and morphological properties of two types of fiber-reinforced composites (carbon fiber and glass fiber). The results of differential scanning calorimetry analysis showed that the quenching process did not affect the relative degree of crystallinity value of CF and GF-reinforced composites [24].

Despite the above-mentioned research, the frictional properties of different carbon fiber reinforced polymer (CFRP) composite materials, which slide against different types of materials under different normal loads and sliding speeds, have not yet been clearly understood.

Nowadays, various composite/composite combinations are widely used for sliding/rolling applications where low friction is required. Although pioneering research by Lancaster and Giltrow for example showed some significant factors affecting the friction and wear of CFRP material [25,26,27], the tribological friction and wear properties of CFRP were not fully understood. The present authors have studied the friction and wear of unidirectionally oriented CFRP in contact with the same material and investigate the tribological behaviors of carbon fiber-reinforced polymer (CFRP) composites under boundary and dry lubrication conditions with respect to sliding direction. 

This work presents:-the results of an investigation on the effect of sliding speed and load on the tribological behavior of two CFRP composite materials tested in a pin-on-disc tribometer under boundary lubrication in the presence of an engine oil with TiO_2_ nanoparticles at 0.075 wt.%. This test configuration is an “open-type”.-a slip pair, referring to self-mated pairs, with the same matrix, reinforcement, and orientation of the fibers on both opposite surfaces (disc and pin).-Experiments were performed at different normal loads (range between 20–80 N) over speeds of 0.4–2.4 m s^−1^ and at room temperature (RT°C).-The duration chosen for the test was 25–26 s on the stick sleep tribometer and then on the pin on the disc, it was 30 min respectively, 360 min for each test.-the friction and wear mechanisms of this tribosystem, the morphological characteristics of the surfaces resulting from wear and the obtained wear residues were analyzed and interpreted.-The influence of the friction time on the coefficient of friction of these composite materials was also examined.

Mainly the content of this paper is organized as follows: structure and properties of the material are presented in Section 2. Section 3 describes the general strategy and related information about the experimental method and device. The experimental equipment used will be discussed separately. Then, in Section 4 the results and discussions are presented. Finally, brief conclusions will be drawn in Section 5.

## 2. Material, Structure and Properties

### 2.1. Material Structure

In the experiments, a carbon-fiber-reinforced polymer (CFRP) composite was selected, made from stacked layers of ex-rayon carbon fibers impregnated with a resin carbon matrix, courtesy of the Company R&G Faserverbundwerkstoffe GmbH, Waldenbuch, Germany, made from high-strength (HT) carbon fiber prepress with transparent epoxy resin matrix in press molding, with a heat deflection temperature 100–105 °C. The chemical characterization and elemental analysis of the materials of the samples used in this work were carried out based on EDS analysis using Oxford EDS—Ultim^®^ Max EDS with AZtecLive software (version life 6.0, Oxford Instruments, Abingdon, UK). 

CFRP composite used for these tribological experimental investigations is composed of 7 layers based on carbon and the type of additives introduced into the binding matrix is the resin carbon matrix. A scanning electron microscope (SEM) image of the CFRP taken along the thickness of the material is shown in Figure 1a,b. Here we can see the 7 layers of the material as well as the thickness of each layer. The material obtained and subjected to the tests has a central layer (layer number 4) of 1 mm and the two marginal layers around 760–800 µm thickness.

The SEM images shown in Figure 2 and Figure 3 indicate that in the 7 composite layers that make up the material under investigation, the fibers are placed once along the surface, then perpendicular to the surface ending with layer 7, which has the same fiber orientation as in the first layer. Based on the analysis with EDS Energy Dispersive X-ray Spectroscopy, this allows us the compositional analysis of a given volume of the sample subjected to tribological investigations and shows us that the composition of the layers is approximately the same, to ensure that each layer corresponds to the parameters on that we want, and the material meets the requirements. 

### 2.2. Nano Tribo-Mechanical Properties

For the evaluation of the nanomechanical and nano tribological properties of the material, tests were performed with the AFM XE-70 (Park System Co., Suwon, South Korea). The tests were repeated 5 times for each analysis and the average results are used. The procedure used for testing involves (i) analysis of surface roughness; (ii) nanoindentation tests to determine the more precise mechanical properties of the elasticity modulus and hardness; (iii) determination of the frictional force between the AFM tip and the tested samples.

The AFM probe to characterize the surface is of the PPP-NCHR type with a thickness of 4 µm, a width of 30 µm, a length of 125 µm, a force constant of 42 N m^−1^, and a 330 kHz resonant frequency. The AFM probe is characterized by a high operational stability and fast scanning capacity. The scan rate during measurement was set at 0.75 Hz, the set point to 2 nm, and the amplitude of oscillations to 20 nm. The scanning area was 40 µm × 40 µm.

For the nanoindentation tests, the probe used is the TD 21,464 with a Berkovich pyramid-shaped tip and a force constant of 156 N m^−1^. Each sample was indented 5 times with a force of 50 µN at 23 °C and the average results were considered and presented in this work (Figure 4). For the hardness, the average value obtained is around (120–140) GPa, and for elastic modulus around (35–45) GPa.

To determine the roughness, the surface scan was performed using the non-contact mode of AFM (NC-AFM). This mode measures surface topography based on atomic forces of attraction between the AFM tip and samples. This method is selected to avoid the scratch and deterioration of the polymeric samples.

The arithmetical mean roughness value (R_a_) is one of the most effective surface roughness commonly adopted in general engineering practice. On the scanning size of 40 µm × 40 µm, several attempts have been made and the resulting average R_a_ roughness is around 20–30 nm (Figure 5).

The rotational deflection of the probe d_z_ is optically recorded while the AFM tip from siliciu is moved laterally across the sample under a controlled normal load (300 nN). The AFM probe is moved in the lateral direction by direct contact between the tip and the investigated composite (Figure 6). The rotational deflections of the AFM cantilever are directly proportional to the friction force between the AFM tip and samples. During the friction force analysis for the investigated composite, a PPP − LFMR type AFM probe optimized for high lateral force sensitivity was used to avoid surface scratching. The geometrical parameters of the AFM probe are the force constant of 0.2 N m^−1^, length 225 μm, width 48 μm, thickness 1 μm, the tip height 15 μm.

The friction force between the AFM probe and the investigated composite, based on the torsion beam theory, can be determined with the relation [20]:(1)$$F_{f}=\frac{dz\cdot r\cdot G\cdot h^{3}\cdot b}{l^{2}\cdot s}=29.49\;nN$$
where d_z_ is the deflection of AFM probe [nm], r—a coefficient equal to 0.33, G—the shear modulus of the AFM cantilever material (G = 50.92 × 10^−3^ N μm^−2^), l—the cantilever length, h—the cantilever thickness, b—the cantilever width, s—tip height of the AFM probe.

The coefficient of the nano friction (µ_nano_) is around 0.098–0.01.

### 2.3. Mechanical Properties

The mechanical properties of the carbon fiber-reinforced polymer composites were determined by tensile testing, according to ISO 527. The tensile specimens were taken from a plate of 5 mm thickness by milling. The tests were conducted on a material-testing machine, Zwick/Roell Z150. The machine is equipped with a biaxial extensometer, able to measure both axial and transverse strain, allowing for the determination of Poisson’s ratio. The stress-strain curves and mechanical properties were determined for five specimens. Figure 7 shows the stress-strain curves, while Table 1 lists the average values of mechanical parameters of the tested material. The material had an average tensile strength of 782 MPa, with a standard deviation of 39.8 MPa. The average strain at tensile strength (and standard deviation) was 1.6% (0.43%). The average values (and standard deviation) for tensile modulus of elasticity and Poisson’s ratio were 43,000 MPa (1050 MPa) and 0.12 (0.062), respectively.

## 3. Experimental Method and Device, General Strategy

The American Society of Lubrication Engineers has published a catalog of friction and wear testing devices describing in detail over 230 different tribometers [7]. Each tribometer has its advantages and disadvantages. For this work, we used the pin on the disc tribometer and a low-velocity linear tribometer.

### 3.1. Pin on Disk Tribometer

The pin on the disk test is shown in the figures below, Figure 8 and Figure 9. The stationary pin is pressed against the rotating disk under the given load. A cylindrical pin (both ends flat) can slide on a horizontal surface, which rotates using the power from a motor. A circular test disc is fixed on a horizontal plate that can rotate and this rotation (rpm) can be varied by an electronic speed control unit.

During the test, the friction force, wear, and temperature were continuously monitored. Discs with an apparent density, at 23 °C, and a range of (1.7–2.0) g cm^−3^ were machined into an outside diameter of 50 mm and a thickness of 5.2 mm, with a tolerance of ±10%. The sliding surfaces (pin and disc) were wet polished with alumina paste, then cleaned with pure ethyl alcohol in a bath and dried by blowing with hot air. The roughness parameter R_a_ values of the polished surfaces were measured to be about 0.30 µm for pin and disc.

The cylindrical pin (6 mm diameter) made of the same material as the disc is mounted in support. The support is fixed by an arm. To measure the frictional force, a force cell was used together with a digital indicator. To obtain the friction coefficient, the measured friction force was divided by the applied normal load. Each experiment was performed for 30 min and 360 min, and after each experiment, the pin and disk were replaced with a new test pair. The experiment was repeated five times to ensure the reliability of the test results, and the average value was taken into account. The corresponding standard deviation is indicated by the error bars, which describe the spread of the results.

After a certain number of cycles, the test is stopped, and an analysis of the surface topography is carried out to determine the amount of wear and the evolution of the surface roughness. At the end of each test, the worn surfaces of the pins and the track surface of the discs were examined by scanning electron microscopy (SEM) and optical microscopy (OM) to provide information on the material removal mechanisms (wear phenomenon).

The measurements of the surfaces were carried out using a Nano Focus-Optical 3D Microscopy, for which the innovative μsurf technology delivers high-resolution 3D measurements of surfaces. It is based on the technology of continuous focus variation with a fixed focal length for the objectives. The scanning objective was 10 x zoom. The surface was analyzed by extracting radial layers of the scanned surface. The roughness R_a_ and R_z_ standards were ISO 4287 and for the R_K_ the measurements were made according to ISO 13565-2 using a double-Gaussian filter of 0.25 mm. The topography layers were measured according to ISO 25178. This information obtained can be used to develop accurate wear models and then proceed to predictive tribological evaluation. Table 2 shows the details of the experimental conditions.

The test was performed in dry and boundary-lubricated (BL) conditions. In the boundary-lubricated case, the oil used was Motul 5100 4 T 10 W-30 engine oil with TiO_2_ Degussa P25 nanoparticles from Sigma–Aldrich. Against the background that TiO_2_ is best suited for many tribological applications (including solid lubricants) due to its excellent tribological behavior, these nanoparticles were chosen for the current investigation. The concentration of TiO_2_ for the engine oil used in the experiments was 0.075 wt.%, which was dispersed into the engine oil, and 0.2 mM Triton X surfactant was added to increase the stability of the nano lubricants. To prevent agglomeration of nanoparticles in the used motor oil and to provide stability for nanoparticles, 0.2 mM Triton X surfactant was used. It plays an important role in mixing the nanoparticles in a way that makes them soluble in the engine oil. The physicochemical properties of engine oil and TiO_2_ nano lubricants are presented in Table 3.

During the experiments, the temperature was monitored with Flir E30 infrared camera, which offers a 160 × 120 IR pixel resolution. The temperature range is between −20 °C to 250 °C and thermal sensitivity of 100 mK or less than 0.1 °C.

### 3.2. Low-Velocity Linear Tribometer

The experiments with low sliding speed (around 110 mm/min) for the material coupling of the carbon fiber-reinforced polymer composite (CFRP) were performed using this tribometer (Figure 10). The tribometer consists of the friction coupling the elements 3 and 4, respectively (contact on flat surfaces), and the helical spring 2 by means of which the element 3 is connected to a fixed point and the vibration damper 1. The CFRP/CFRP test set is loaded with a different normal load from (6 to 16 N) at a slip speed of 110 mm·min^−1^ under boundary lubrication conditions over 50 mm for 28 s. During the period when element 3 is moving, the minimum and maximum values of the friction forces are continuously recorded, based on which the static and dynamic friction coefficient is then determined.

## 4. Experimental Results

In general, two types of friction response, namely steady-state and stick-slip, can be observed during sliding. During the stick-slip period, the friction force does not remain constant, but oscillates significantly depending on the distance or the sliding time. During the stick phase, the friction force increases to a critical value. Once the critical force has been reached, the static friction is overcome and sliding occurs at the interface of the surfaces in contact and the energy is released so that the frictional force decreases. This stick-slip phenomenon occurs when the static friction coefficient is higher than the kinetic friction coefficient. Bowden and Tabor (1950) suggested that static friction is greater than kinetic friction due to molecular bonding between the surfaces. When a countersurface slides against a CFRP surface, both normal load (F_n_) and tangential force (F_f_) are supported by fibers and matrix, so that the friction coefficient COF can be given by:(2)$$\mu =\frac{F_{f}}{F_{n}}=\frac{F_{fibre}+F_{matrix}}{F_{n\;fibre}+F_{n\;matrix}}$$

First, we studied the static and kinetic friction for the pairs of CFRP composite under boundary-lubricated conditions (using the oil added above) as a function of normal loads based on a low-velocity linear tribometer. It was observed that both static and kinetic coefficients of friction were slightly lower under boundary lubrication. It was noted for this very smooth surface that the value of the static and kinetic coefficient of friction decreases with increasing normal loads. More precisely, we can say that the static and dynamic coefficient of friction decreased slightly with increasing normal load on the CFRP composite pairs with polished discs. Figure 11 presents this variation where we can observe that for 6 N loaded the COF_static_ is around 0.45 and COF_kinematic_ ~ 0.4 and for 16 N loaded COF_static_ ~ 0.28 and COF_kinematic_ ~ 0.24.

The stick-slip tests were repeated several times and the recordings of the minimum and maximum frictional forces were made continuously from the program resulting in the values of the static and dynamic coefficient of friction (Figure 12). The test gives us an image of the behavior of this CFRP in conditions of boundary lubrication and very low speeds.

Using the pin on disc tribometer, the variation of the friction coefficient was followed in the test conditions specified above under the action of various loading forces. The temperature in the contact area was also monitored throughout the tests. The tests were performed for 30 min and then the experiments were extended to 360 min to determine the behavior of this pair of materials over time.

As can be seen from the figures plotted below (Figure 13, Figure 14, Figure 15 and Figure 16), at the beginning of the test, the measured coefficient of friction (COF) is slightly higher and then decreases progressively. This behavior is typical for friction measurements and is attributed to the run-in phenomenon. During running-in, the topography of the surface changes, and the chemical reactions on the surface take place until the tribological system reaches equilibrium. This equilibrium COF value is usually reported. In the case of the material used, the surface changes are minor, and the values of the friction coefficients stabilize in a few minutes from the beginning of the test. We can observe that for the force between 20–60 N the COF stabilizes after 10 min of tests and after 20 min you can see even a slight decrease in the COF (Figure 13). With the increase of the testing period to 360 min, we can say that already after 50 min of testing the COF is stabilized (Figure 15).

On the other hand, with the increase in the force to 80 N after 10 min of tests, a progressive increase in the COF can be observed. After about 180 min of testing, the local temperature continuously rises above 100 °C and the friction coupling from CFRP has ceased (Figure 16).

A similar variation in the COF also appears due to the change in the sliding speed and is presented in Figure 17; namely, at the beginning of the test, the measured coefficient of friction (COF) is slightly higher and then decreases progressively. The decrease is more pronounced for the sliding speed of 2.4 m·s^−1^. As for the local temperature, due to the pin on the disc fault after 15–20 min of testing in all situations of speed, variation has stabilized (Figure 18), and the temperature does not exceed 55 °C.

Wear that occurs under equilibrium conditions during repetitive sliding in the same way on CFRP composite against the bottom specimen from the same material is probably a consequence of localized adhesion to the roughness contacts between the composite on composite. Wear rates for each material were measured based on interpreting the roughness parameters of the disc surface. This variation due to roughness has been observed elsewhere for Perspex (Lancaster 1963) [17] and is now a general feature for other composite materials. In [17], it is presented that fiber orientation might be a significant factor in friction and wear. It is also mentioned that the lowest wear is obtained when the fibers are perpendicular to the sliding surface, and the highest when the fibers lie parallel to the surface and in the direction of motion [17]. 

For our experiments, to examine friction and wear, the closing layers of the multilayer structures of CFRP composite are made of linearly oriented fibers. With this sliding arrangement, the effect of the orientation was particularly marked and Figure 19, Figure 20, Figure 21, Figure 22 and Figure 23 show the traces of the profilometer along the wear tracks on the disc under the chosen test conditions.

Examination of the images in Figure 19, Figure 20, Figure 21, Figure 22 and Figure 23 suggests that the composite exposed during slipping may withstand the applied loads. During the tests, two series of experiments were conducted. First, the friction and wear rate of the prepared composite were measured at different loads, and the results for this are shown in the figures below. The values presented were obtained for a maximum speed of 217 rpm of the disc to minimize the effects due to the frictional heating of the composite samples in contact at higher speeds. The second series of experiments involved a determination of the variation of the wear rate as a function of the sliding speeds. Microscopic examination, along with surface profiles such as those in Figure 19, Figure 20, Figure 21, Figure 22 and Figure 23, confirmed that the composite surface was in fact appreciably smoothed, and material transfer from one surface to another was almost completely absent.

In normal sliding, every CFRP suffered a seizure after several kilometers of sliding distance, so these data are not given in the figures below. The microscopic examination, along with profiles for the case where the peripheral speed was changed, was performed with the same method as in the case of force variation, the results being included in Table 4.

For a more rigorous interpretation of the wear investigated, the specific changes in worn surfaces, we proceeded to the evaluation of the surface roughness parameters. The measurements were carried out on a sampling length of 0.8 mm and evaluation length of 4 mm using a Gaussian filter according to ISO GPS standards recommendation based on INSIZE ISR-C300 roughness tester (Figure 24).

For each sample, three measurements were carried out in three different sections and the mean values of the obtained parameters are presented in Table 4.

Slight changes were determined on the surface roughness, only on the sample which was tested for 1 h 30 min at F_n_ = 80 N—seizure, the total height of the profile R_t_, the sum of the height of the largest profile peak height, and the largest profile valley depth, within the evaluation length, was in some cases higher than 40 µm. Regarding the other parameter, R_a_–arithmetic mean deviation of the absolute ordinate values within a sampling length, very small variations of those values were registered.

The values of the measurements on the original surface are a bit higher than the values obtained after the experiments were conducted, meaning that the irregularities were compressed during the wearing procedure.

The overall measured values show there were no significant changes regarding the roughness parameters.

## 5. Discussion

To avoid the occurrence of severe seizure of motion components exposed to mixed and boundary lubrication, e.g., in engine and transmission systems, replacing metal–metal friction pairs with polymer–polymer pairs provide a potential solution. We used boundary lubrication in the idea that these material couplings are used in mechanical systems that have frequent stops and starts and where protection against sudden shocks is critical. When factors such as load and speed do not make it possible to obtain a continuous film of lubricant, oils will be infused with additives for anti-wear and extreme pressure to provide additional protection for surfaces.

Because in the research carried out on materials, metal coatings, and lubricants, standardized tests are used to improve the reproducibility of the results, and for the simplicity of the tests, we used the pin-on-disk test method in these experiments. The pin on disc, pin on ring, pin on flat, and ball on flat tests are all based on the same principle, with differences in the geometry of a particular tribometer and the pin. Thus, the data collected through standardized tests can be compared with those obtained in other laboratories.

In the presence of the lubricant, the polymeric materials interact with the oil in several ways. The absorption of oil molecules onto the surface is primarily a diffusion-controlled process and depends on time, temperature, stress, the ambient environment as well as the molecular configuration of both the oil and the polymer. This swelling effect of the surface layers leads to a decrease in the shear strength of the composite, thus reducing its friction.

Due to the additive oil as a lubricating medium, it works as a cooling agent, contributing to the mitigation of the thermal and mechanical effects induced by friction, which leads to the improvement of the softening and plastic deformation of the polymer matrix. In addition, due to the boundary lubrication (BL) effect of the oil-absorbed layer, it makes the wear of the composite decrease greatly compared to that under dry sliding. Moreover, the oil could wash the surface, in other words, the effect of the oil is to remove the debris from the friction region, reduce the abrasive wear, and finally increase the polishing effect of the carbon fiber on the counterface of the friction sample, reducing the surface roughness. Therefore, a reduced wear rate was attained in this test condition.

Additive oil molecules can easily fill the cavities on the counterfaces, which means that the wear debris will be excluded from these cavities by the hydraulic effect.

Otherwise, the lubricant removes debris from the friction zone and prevents wear debris from accumulating on the surface. So, we cannot highlight the appearance of the transfer film on the counterface under lubrication conditions.

During the running-in period, the topography of the surface changes, and the chemical reactions on the surface take place until the tribological system reaches equilibrium. This equilibrium of the COF value is usually reported.

Previous research indicated also in the introduction section shows that under dry friction conditions, the friction coefficient of the PPS–PPS combination is the highest value of all the combinations and is about 0.8. The friction coefficients of PA66–PA66 and PTFE–PTFE combinations, which are about 0.6 and 0.3, respectively, are lower than that of the PPS–PPS combination. The investigation was conducted approximately on the same conditions and same devices. Our results under boundary lubrication vary between 0.08 to 0.18.

As for the oil-lubricated conditions, the friction coefficients of all combinations in previous research were below 0.1, indicating that the friction coefficients of all combinations can be reduced by one order of magnitude compared to those under dry friction conditions. The presented research reported the COF value as ranging between 0.08 and 0.04.

In the previous research the specific wear rates of PTFE, PA66 and PPS discs under dry friction and oil-lubricated conditions found that the wear of the polymer discs is in the decreasing order of PTFE > PA66 > PPS under both dry friction and oil-lubricated conditions. It is observed that the wears of PTFE and PPS are lower under oil lubrication than those under dry friction, while the wear of PA66 is increased by adding oil on the sliding surface. For instance, the wear rate was presented to be in dry conditions between 1 × 10^−4^ mm^3^ N^−1^ m^−1^ up to 14 × 10^−4^ mm^3^ N^−1^ m^−1^. Under oil lubrication, previous research reported values between 0.1 × 10^−4^ mm^3^ N^−1^ m^−1^ up to 6 × 10^−4^ mm^3^ N^−1^ m^−1^.

Our samples were evaluated under microscopic examination, which, along with surface profiles (results are presented in Figure 19, Figure 20, Figure 21, Figure 22 and Figure 23), confirmed that the composite surface was in fact appreciably smoothed, and material transfer from one surface to another was almost completely absent. Our results and the literature reported that the friction coefficients of all sliding combinations decrease with the increasing applied load for dry friction, while the friction coefficients were independent of applied loads for oil-lubricated conditions. We can mention that our results in BL show that the COF varied between 0.08 and 0.2.

From the literature, the friction coefficient of PA66–PA66 combination under dry friction condition increases to the maximum value when the sliding speed is 0.4 m/s; after that, the friction coefficient decreases slightly with the increasing sliding speed; the friction coefficient of PTFE–PTFE under dry sliding is independent of sliding speed. As for oil-lubricated conditions, the friction coefficients of all sliding combinations are independent of sliding speed. A similar variation occurs in our tests with the COF due to the change in the sliding speed and is presented in Figure 17; namely, at the beginning of the test, the measured coefficient of friction (COF) is slightly higher and then decreases progressively. The decrease is more pronounced for the sliding speed of 2.4 m·s^−1^.

Clearly, the relationships between the mechanical properties of the composite and their wear characteristics remain somewhat unclear and not fully understood. Further investigations are needed to determine the most effective additives and their proportions in the mass of composite material to obtain an optimal compromise between mechanical and tribological properties. Furthermore, an important aspect of future research is the examination of how the fibers are oriented and how this contributes to the modification of the mechanical and tribological properties of the composite material.

In comparison with a vast number of investigations under dry friction conditions, much fewer studies were performed with oil lubrication. Even though pioneering research on oil-lubricated polymeric materials was carried out, systematic formulation theories of high-performance composites for applications subjected to oil lubrication conditions are still lacking.

## 6. Conclusions

From the above, the following conclusions can be drawn:-The applied load as well as the sliding speed have an important contribution to controlling the friction and wear behavior of the composite materials.-The worn surface exposed to abrasive wear was predominant in the tested samples. These tests result in two important aspects:○firstly, the disc is subject to the effects of wear due to intermittent loading, unlike the stationary sample or the pin; ○secondly, it is more difficult to analyze the resulting wear debris that tends to be driven away by the centrifugal forces that occur during testing. 
-The wear values of the disc are comparatively higher than those of the pin, so we focused the analysis only on it.-The mechanism of reducing the wear of polymers by carbon fiber reinforcement seems to involve two factors.
○First, the fibers are exposed to the sliding surface and support part of the applied load; ○Second, the fibers smooth the surface of the counterface and reduce the stresses that appear on the contacts between the roughness peaks.
-A coefficient of friction of 0.25–0.35 for stick-slip phenomena and 0.08–0.12 under boundary lubrication is in fact in good agreement with other research works.-The coefficient of friction and the wear limit are among the lowest observed; the elastic modulus and hardness are both considerably higher.

## Figures and Tables

**Figure 1 polymers-14-03716-f001:**
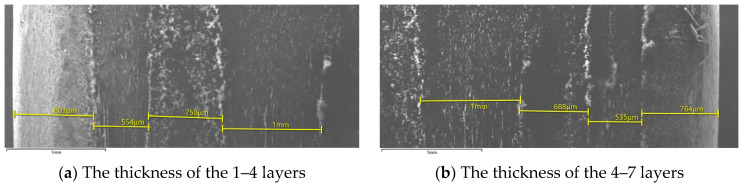
SEM images of the layers that make up the tested composite material.

**Figure 2 polymers-14-03716-f002:**
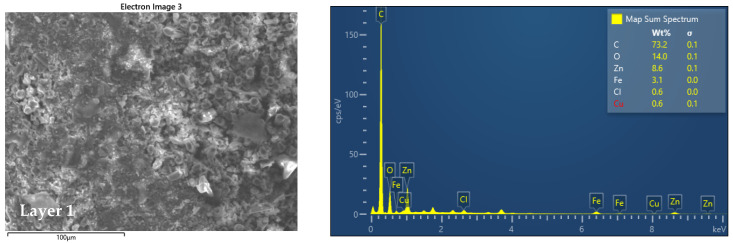
SEM images and EDS analysis of layer 1.

**Figure 3 polymers-14-03716-f003:**
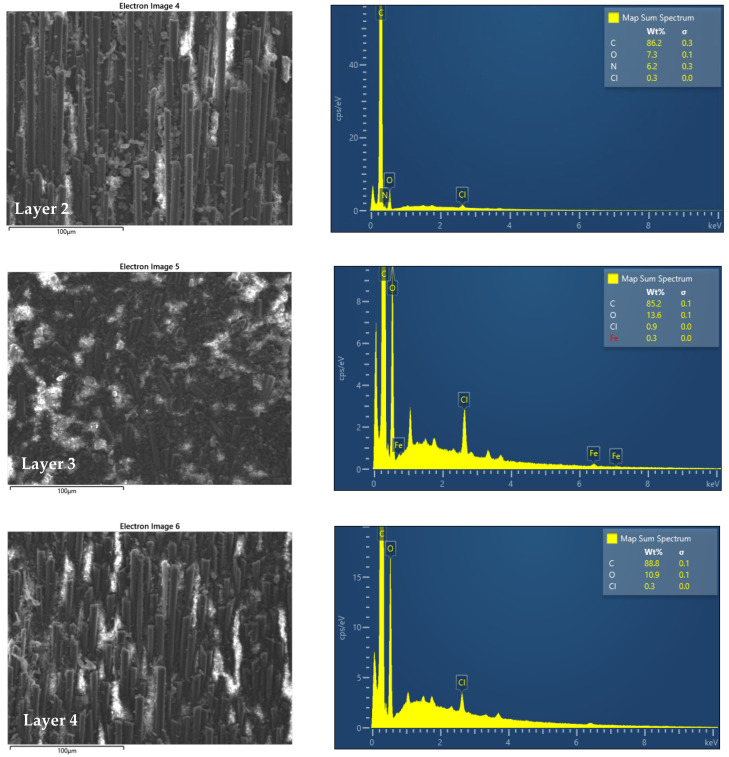
SEM images and EDS analysis of layers 2–7.

**Figure 4 polymers-14-03716-f004:**
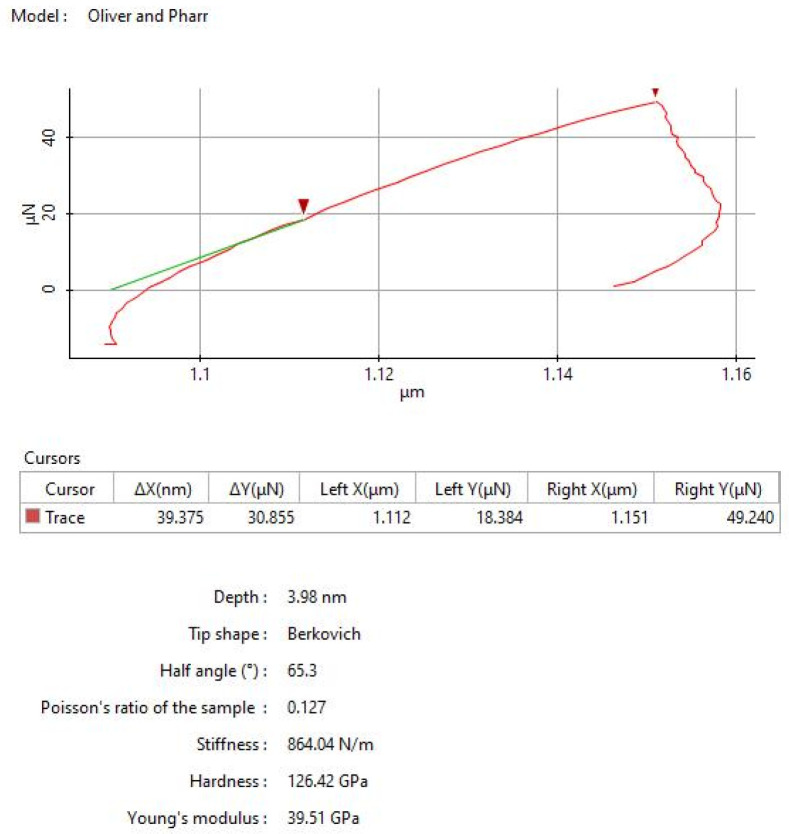
Nanoindentation of CFRP composite (50 µN).

**Figure 5 polymers-14-03716-f005:**
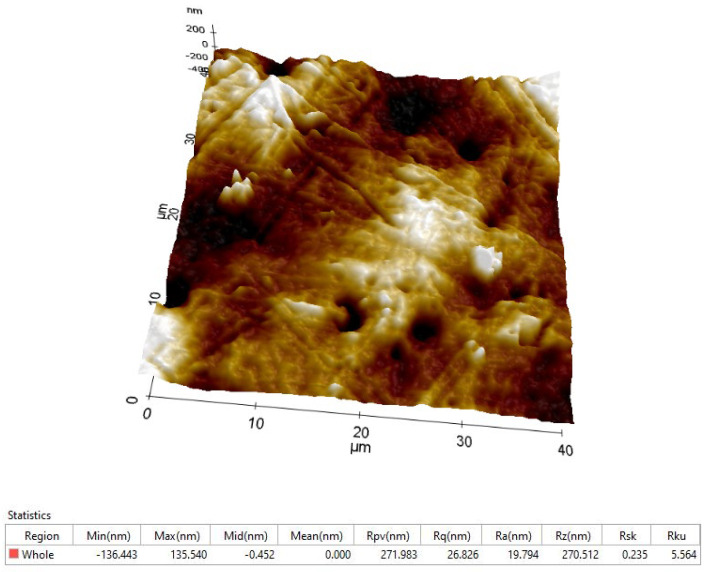
Topography (3D image) and the roughness parameters of CFRP composite.

**Figure 6 polymers-14-03716-f006:**
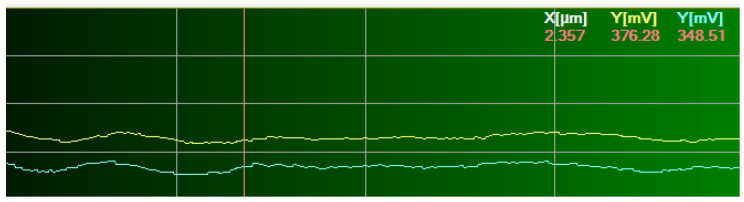
Optical monitoring of the rotational deflection (d_z_) of the AFM probe during sliding.

**Figure 7 polymers-14-03716-f007:**
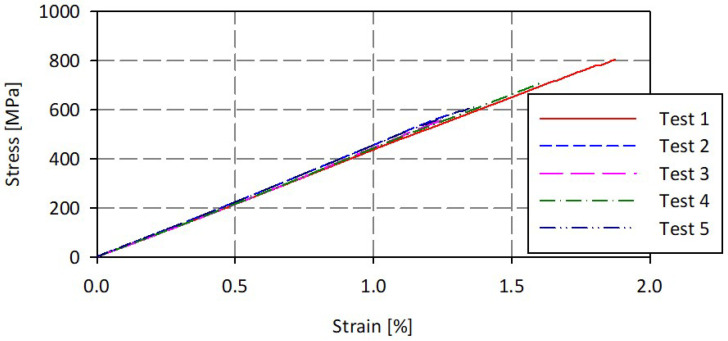
Stress-strain curves.

**Figure 8 polymers-14-03716-f008:**
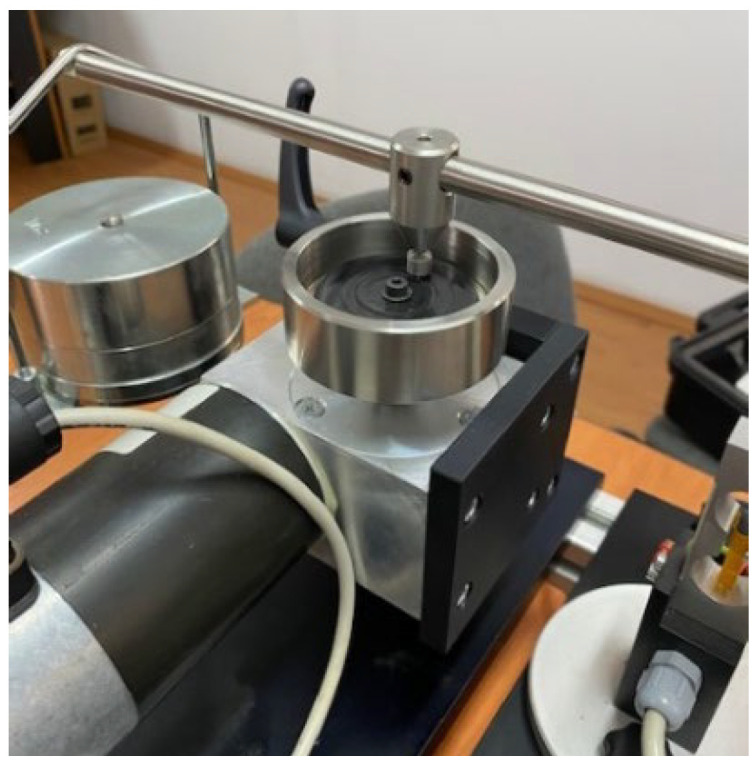
Pin on disk tribometer.

**Figure 9 polymers-14-03716-f009:**
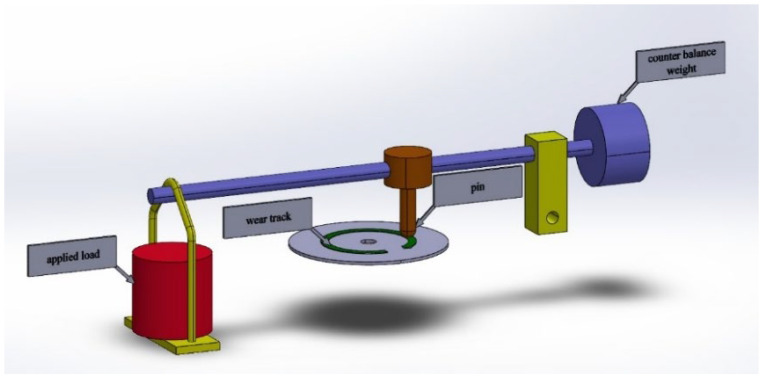
Schematic representation of a pin on disk tribometer.

**Figure 10 polymers-14-03716-f010:**
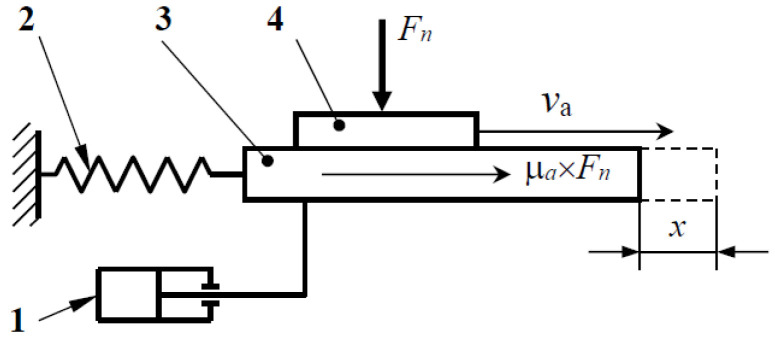
Schematic representation of a low-velocity linear tribometer.

**Figure 11 polymers-14-03716-f011:**
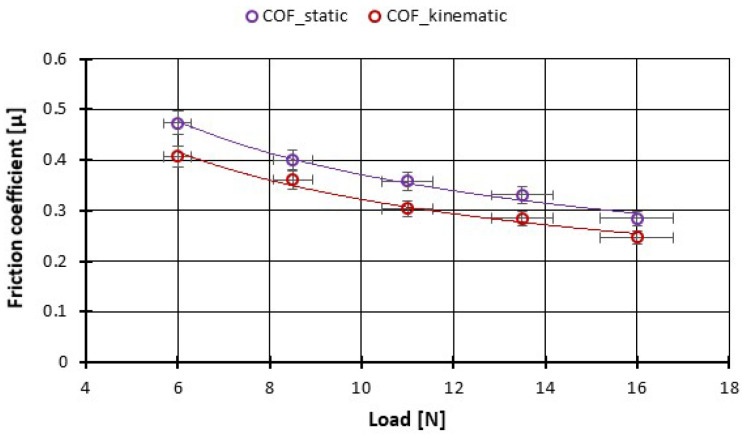
The variation of the friction coefficient of CFRP composite against the bottom specimens versus load at low speed (Stick-slip phenomena).

**Figure 12 polymers-14-03716-f012:**
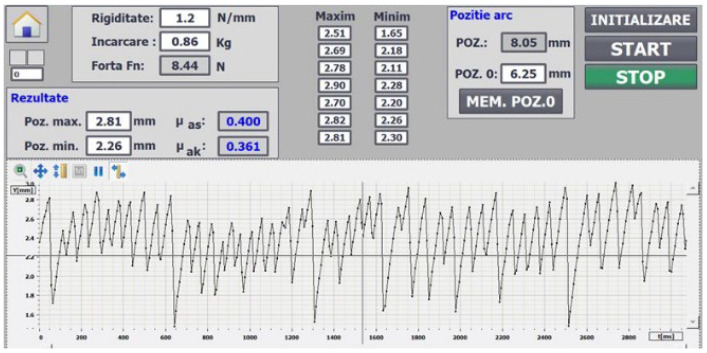
Variation of the friction force in relation to time.

**Figure 13 polymers-14-03716-f013:**
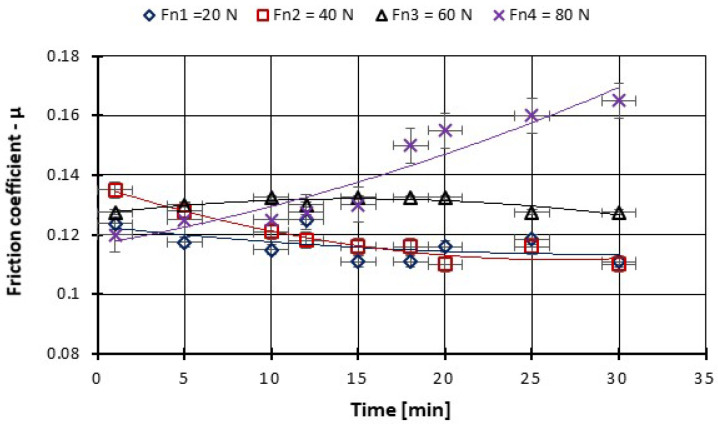
The variation in the friction coefficient of CFRP composite against the bottom specimen versus load at test time (30 min).

**Figure 14 polymers-14-03716-f014:**
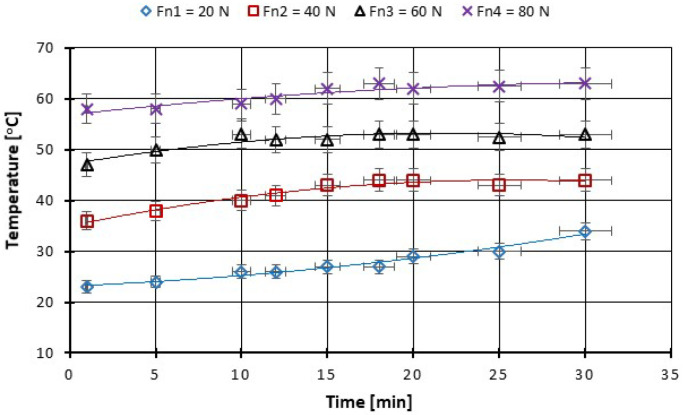
The change in boundary oil temperature vs. load, at test time (30 min).

**Figure 15 polymers-14-03716-f015:**
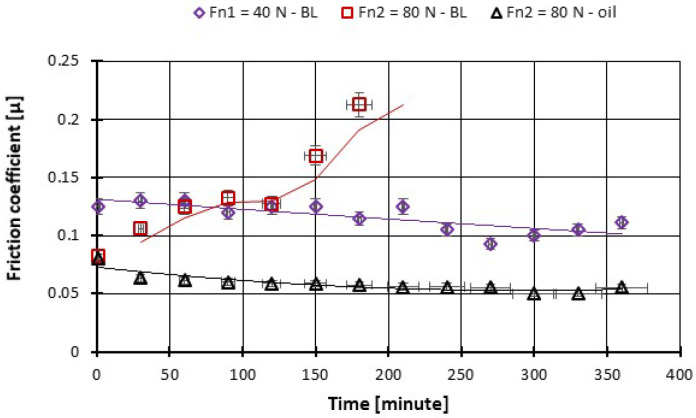
The variation in the friction coefficient of CFRP composite against the bottom specimen versus load at test time (360 min).

**Figure 16 polymers-14-03716-f016:**
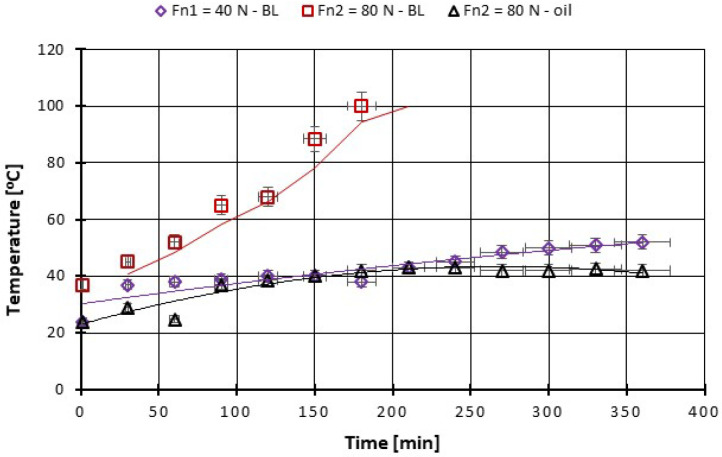
The change in boundary oil temperature vs. load, at test time (360 min).

**Figure 17 polymers-14-03716-f017:**
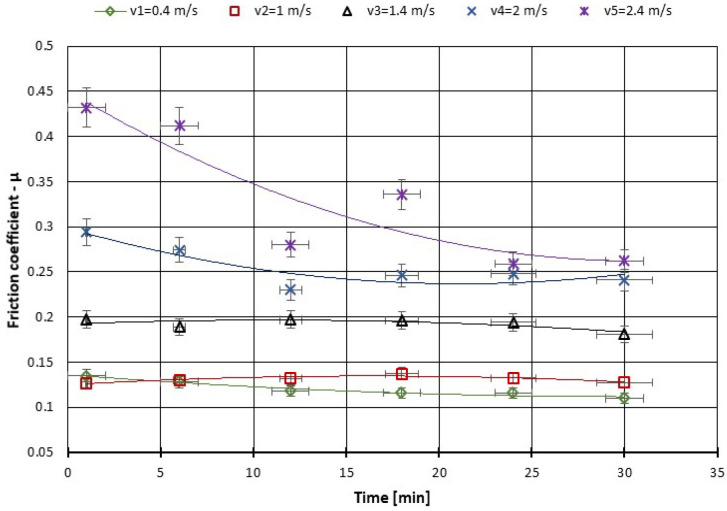
The variation of the friction coefficient of CFRP composite against the bottom specimen versus peripheral speed at test time (30 min).

**Figure 18 polymers-14-03716-f018:**
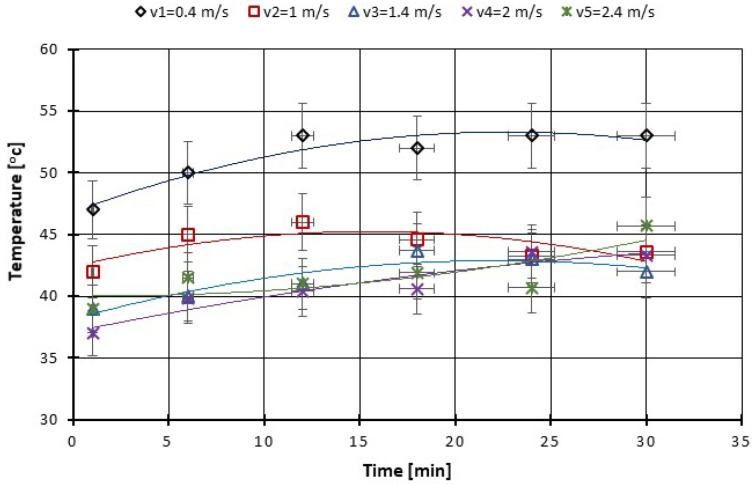
The change of boundary oil temperature versus peripheral speed at test time (30 min).

**Figure 19 polymers-14-03716-f019:**
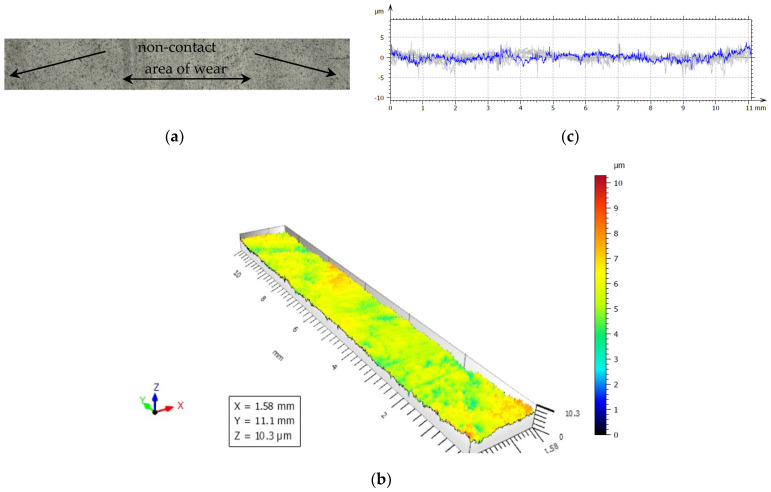
(**a**) The worn and unworn surfaces morphologies of CFRP, (**b**) 3—D profiles of wear scars on the CFRP after friction experiment boundary lubricated under F_n_ = 20 N, duration 30 min. (**c**) Height variations of wear marks.

**Figure 20 polymers-14-03716-f020:**
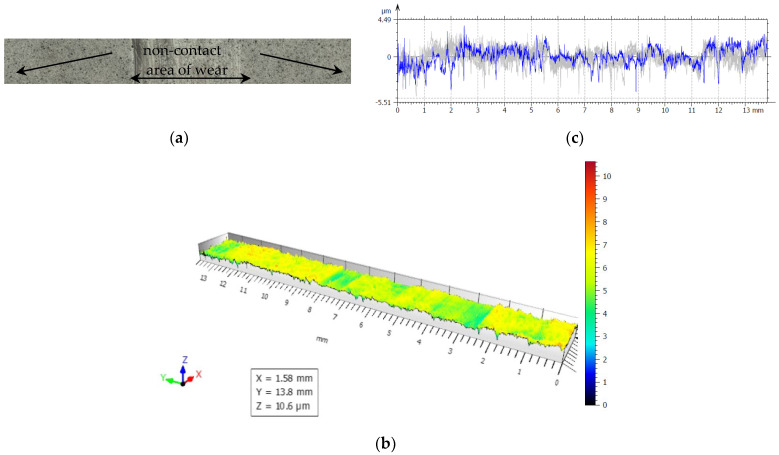
(**a**) The worn and unworn surfaces morphologies of CFRP, (**b**) 3—D profiles of wear scars on the CFRP after friction experiment boundary lubricated under F_n_ = 40 N, duration 30 min. (**c**) Height variations of wear marks.

**Figure 21 polymers-14-03716-f021:**
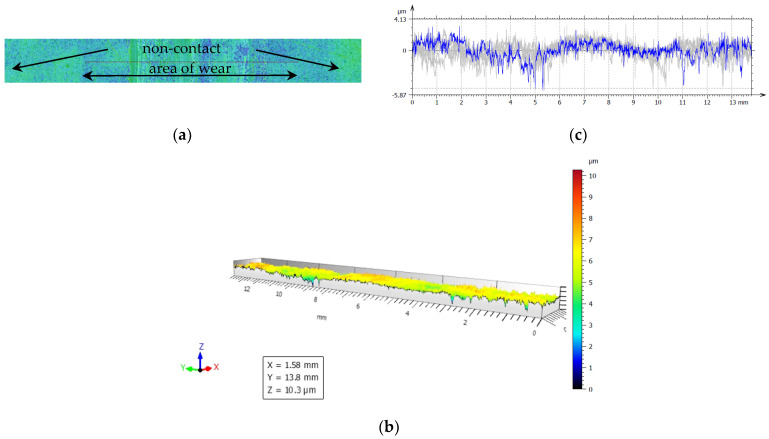
(**a**) The worn and unworn surfaces morphologies of CFRP, (**b**) 3—D profiles of wear scars on the CFRP after friction experiment boundary lubricated under F_n_ = 40 N, duration 360 min, (**c**) Height variations of wear marks.

**Figure 22 polymers-14-03716-f022:**
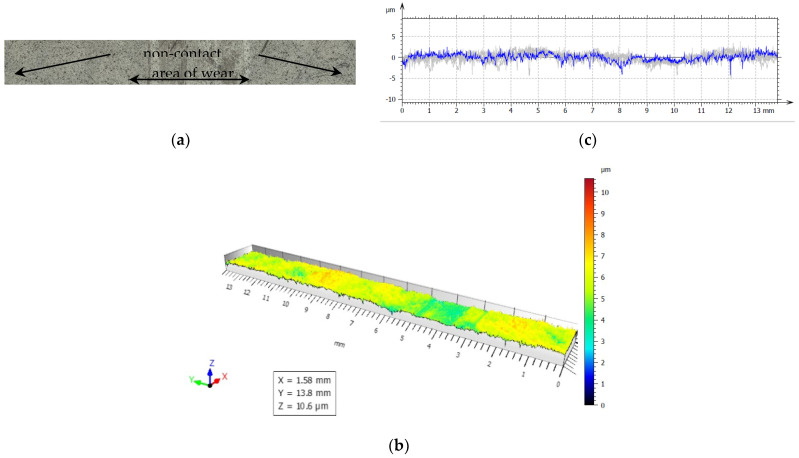
(**a**) The worn and unworn surfaces morphologies of CFRP, (**b**) 3—D profiles of wear scars on the CFRP after friction experiment boundary lubricated under F_n_ = 60 N, duration 30 min, (**c**) Height variations of wear marks.

**Figure 23 polymers-14-03716-f023:**
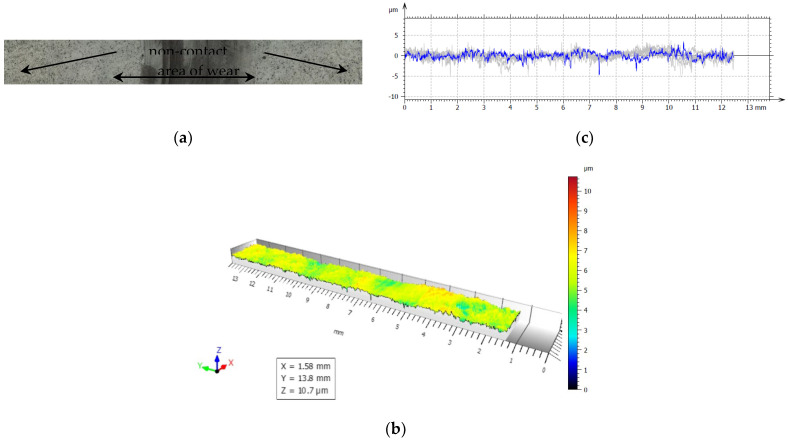
(**a**) The worn and unworn surfaces morphologies of CFRP, (**b**) 3—D profiles of wear scars on the CFRP after friction experiment boundary lubricated under F_n_ = 80 N, duration 30 min, (**c**) Height variations of wear marks.

**Figure 24 polymers-14-03716-f024:**
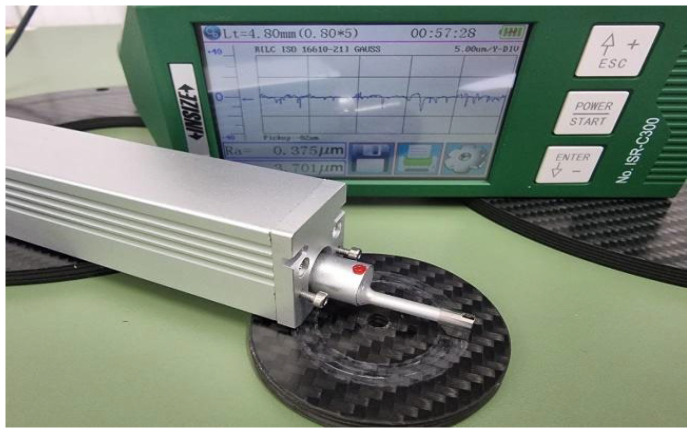
Experimental set-up for roughness measurements using the INSIZE ISR—C300 device.

**Table 1 polymers-14-03716-t001:** Tensile properties of the carbon fiber-reinforced polymer composites.

Parameters	Mean Value	Standard Deviation
Modulus of elasticity [MPa]	43,000	1050
Tensile strength [MPa]	782	39.8
Strain at tensile strength [%]	1.6	0.43
Poisson’s ratio	0.12	0.062

**Table 2 polymers-14-03716-t002:** Experimental conditions.

Parameters	Operating Conditions
Normal load	20, 40, 60, 80 N
Sliding velocity	0.4, 1.0, 1.4, 2.0, 2.4 m s^−1^
Rotating speed	217 (±3) rpm
Relative humidity	45 (±5)%
Starting temperature (RT)	23 (±2) °C
Duration of rubbing	30/360 min
Surface conditions	Dry/boundary lubrication (BL)
Disc/pin material	Carbon fiber reinforced polymer (CFRP) composite
Average surface roughness R_a_	µm

**Table 3 polymers-14-03716-t003:** The physicochemical properties of engine oil and TiO_2_ nano lubricant [5].

	Density at 15 °C [kg m^−3^]	Kinematic Viscosity at 40 °C [mm^2^ s^−1^]	Kinematic Viscosity at 100 °C [mm^2^ s^−1^]	Viscosity Index
pure base oil	862	89.7	13.6	154
0.075 wt.% of TiO_2_	864	92.3	14.2	158

**Table 4 polymers-14-03716-t004:** Measured roughness parameters.

	R_a_ [µm]	R_q_ [µm]	R_p_ [µm]	R_3z_ [µm]	R_z_ [µm]	R_t_ [µm]	R_v_ [µm]	R_3y_ [µm]	R_zjis_ [µm]	R_y_ [µm]
1 h 30 min-BL F_n_ = 80 N (seizure)	2.08	2.91	5.68	3.79	14.95	40.25	9.26	5.24	5.69	14.91
Dry lubrication F_n_ = 80 N (seizure)	0.46	0.62	1.73	2.19	4.06	7.01	2.32	2.70	2.46	6.67
5h-BL F_n_ = 40 N	0.36	0.49	0.88	1.39	3.03	4.91	2.14	1.98	1.65	4.67
5 h-immersed in oil (fluid lubrication) F_n_ = 80 N	0.33	0.45	0.94	1.37	2.83	3.95	1.89	2.12	1.55	3.76
30 min-BL F_n_ = 20 N	0.36	0.52	0.90	1.49	3.47	6.89	2.56	2.20	1.80	6.55
30 min-BL F_n_ = 40 N	0.32	0.43	0.86	1.19	2.62	3.87	1.76	1.50	1.41	3.68
30 min-BL F_n_ = 60 N	0.50	0.71	1.13	1.64	4.20	6.98	3.07	2.71	2.00	6.64
30 min-BL F_n_ = 80 N	0.45	0.67	0.93	1.18	3.93	7.82	3.00	1.52	1.72	7.43
30 min-BL F_n_ = 40 N v_al_ =1 m∙s^−1^	0.30	0.43	0.98	1.36	2.70	5.07	1.71	2.42	1.55	4.82
30 min-BL F_n_ = 40 N v_al_ =1.4 m∙s^−1^	0.53	0.72	1.94	2.84	4.16	6.67	2.30	4.86	2.94	6.34
30 min-BL F_n_ = 40 N v_al_ =2 m∙s^−1^	0.31	0.44	0.93	1.28	2.83	4.59	1.90	2.20	1.46	4.36
30 min-BL F_n_ = 40 N v_al_ =2.4 m∙s^−1^	0.37	0.51	1.23	1.53	3.42	5.52	2.19	2.45	1.86	5.25

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
