# Peer review of "Experimental Investigation of the Tribological Behaviors of Carbon Fiber Reinforced Polymer Composites under Boundary Lubrication"

_polymers, 2022, doi:10.3390/polym14183716_

Round 1

Reviewer 1 Report

The article titled “Experimental investigation of the tribological behaviors of carbon fiber reinforced polymer composites under boundary lubrication” is presented in a scientific way. There are some suggestions to improve the quality of the article.

1 – The introduction section is too long. It is suggested to make it brief.

2 – There are many sentences without references in the introduction section. Please add references.

3 - Please write clearly the objective and significance of your work in the last paragraph of the introduction section.

4 - Add a discussion section before the conclusion regarding the comparison of this study with previous studies and the practical implementation of the current study.

5 - Please make bullet points in the conclusion, and make it brief.

6 - References are too less and too old. Add at least 30 from recent years.

Author Response

Manuscript Polymers - Submission Polymers-1866217

Dear Allen Tian

Special Issue Editor

Thank you for giving me the opportunity to submit a revised draft of our manuscript entitled "Experimental investigation of the tribological behaviors of carbon fiber reinforced polymer composites under boundary lubrication" to Polymers.

We appreciate the time and effort that you and the reviewers have dedicated to providing your valuable feedback on our manuscript. We are grateful to the reviewers for their insightful comments on our paper. We have been able to incorporate changes to reflect most of the suggestions provided by the reviewers. We have highlighted the changes within the manuscript.

Here is a point-by-point response to the reviewers’ comments and concerns.

Comments from Reviewer 1

Reviewer 1

The article titled “Experimental investigation of the tribological behaviors of carbon fiber reinforced polymer composites under boundary lubrication” is presented in a scientific way. There are some suggestions to improve the quality of the article.

Comment 1: The introduction section is too long. It is suggested to make it brief.

Response: Thank you for pointing this out. We have tried to respond to your requests, and we have improved the introduction. The introductory part, as can be seen in the revised manuscript, has been updated according to the requirements of the reviewer

Comment 2: There are many sentences without references in the introduction section. Please add references.

Response: Along with the rethinking of the introduction chapter, the necessary bibliography was also added, so that the entire introduction chapter makes clear reference to the bibliography that was discussed.

Comment 3: Please write clearly the objective and significance of your work in the last paragraph of the introduction section.

Thank you for the suggestion and for this, we have decided to present in the last part of the introduction the important objectives of the work.

Comment 4: Add a discussion section before the conclusion regarding the comparison of this study with previous studies and the practical implementation of the current study.

Thank you for the suggestion. We carefully analyzed this suggestion and we decided that Section 4 entitled Experimental results and discussion, to became Experimental results and to added the section 5: Discussions so that the Conclusions become section 6

Comment 5: Please make bullet points in the conclusion, and make it brief.

Thank you for pointing this out. We have tried to respond to your requests, and we have pointed the conclusions.

Comment 6: References are too less and too old. Add at least 30 from recent years.

Thanks for the suggestion for this I have re-analyzed the bibliography and improved the article with newer references. Throughout the revised manuscript, we have considered these suggestions.

Finally, we can say that we have tried to incorporate all your suggestions throughout the manuscript. All the changes were made using Track Changes.

We hope the revised manuscript will better suit the Polymers but are happy to consider further revisions, and we thank you for your continued interest in our research.

We look forward to hearing from you in due time regarding our submission and to responding to any further questions and comments you may have.

Sincerely,

Corina Birleanu – first author

Marius Pustan and Grigore Pop – corresponding authors

31.08.2022

Reviewer 2 Report

This research topic is interesting, however, technical writing and structure of the manuscript need significant improvement.  Results section need more discussion in depth.

1. in page 3, line 121, what is the results of Sinmazcelik's study?

2. there are several typos in page 3, line 128-131

3. the first 3 paragraphs in section 2.1 are just duplication of introduction;  sentences in the 4th paragraph need rephrase. there is no a. and b in figure 1.

4. what is the purpose of elemental analysis of each layers, Figure 2 and 3?

5. what is BL in Figure 15?

6. there is no study on the effect of TiO2 in section 4?

Author Response

Manuscript Polymers - Submission Polymers-1866217

Dear Allen Tian

Special Issue Editor

Thank you for giving me the opportunity to submit a revised draft of our manuscript entitled "Experimental investigation of the tribological behaviors of carbon fiber reinforced polymer composites under boundary lubrication" to Polymers.

We appreciate the time and effort that you and the reviewers have dedicated to providing your valuable feedback on our manuscript. We are grateful to the reviewers for their insightful comments on our paper. We have been able to incorporate changes to reflect most of the suggestions provided by the reviewers. We have highlighted the changes within the manuscript.

Here is a point-by-point response to the reviewers’ comments and concerns.

Comments from Reviewer 2

Reviewer 2

This research topic is interesting, however, technical writing and structure of the manuscript need significant improvement. Results section need more discussion in depth.

General Response:

Thank you for your point of view and in this regard, the entire introduction section has been revised. Along with the rethinking of the introduction chapter, the necessary bibliography was also added, so that the entire introductory chapter makes clear reference to the bibliography that was discussed. We have decided to present in the last part of the introduction the important objectives of the work. We carefully analyzed and decided that Section 4 entitled Experimental results and discussion, to become Experimental results and to add the section 5: Discussions so that the Conclusions become section 6. Also, we have pointed out the conclusions.

Comment 1: in page 3, line 121, what is the results of Sinmazcelik's study?

Response: Thank you for the suggestion and for this, we introduced the following addition “They evaluated the effects of the microstructural changes, resulting from the thermal aging process, on the tribological, thermal, thermomechanical, physical and morphological properties of two types of fiber-reinforced composites (carbon fiber and glass fiber). The results of differential scanning calorimetry analysis showed that the quenching process did not affect the relative degree of crystallinity value of CF and GF reinforced composites [16].”

Comment 2: there are several typos in page 3, line 128-131

Response: Thank you for pointing this out. We have tried to respond to your requests, and we have improved the introduction. The introductory part, as can be seen in the revised manuscript, has been updated according to the requirements of the reviewer, and the mistakes were corrected, including the typographical ones

Comment 3: the first 3 paragraphs in section 2.1 are just duplications of introduction;  sentences in the 4th paragraph need rephrase. there is no a. and b in figure 1.

Response: Thank you for the suggestion and for this, we have decided to remove from the manuscript the 3 phrases that are repeated, leaving them in the introductory section.

Also, in figure 1, (a) and (b) were highlighted.

Comment 4: what is the purpose of elemental analysis of each layers, Figure 2 and 3?

Response: We wanted to show through the SEM images shown in figures 2 and 3 that in the 7 composite layers that make up the material under investigation, the fibers are placed once along the surface, then perpendicular to the surface ending with layer 7, which has the same fiber orientation as in the first layer. From the point of view of the analysis with EDS Energy Dispersive X-ray Spectroscopy, this allows us the compositional analysis of a given volume of the sample subjected to tribological investigations and shows us that the composition of the layers is approximately the same, to ensure that each layer corresponds to the parameters on that we want and the material meets the requirements.

Comment 5: what is BL in Figure 15?

Response: Refers to boundary lubricated (BL) conditions, which were introduced for the first time on page 10 line 298.

Comment 6: there is no study on the effect of TiO2 in section 4?

Response: No references were made here about the effect of the nanoparticle on the oil. It was not so much a point of interest here. This has been analyzed and can be found in other works of the collective, such as the work “(Effect of TiO2 nanoparticles on the tribological properties of lubricating oil: an experimental investigation. Scientific Reports, 12(1), 1-17. 2022).”

Also, the work team is interested in these aspects regarding the influence of additives with nanoparticles of the lubricant that will be used in the tribological tests of polymer-based samples in future research.

Finally, we can say that we have tried to incorporate all your suggestions throughout the manuscript. All the changes were made using Track Changes.

We hope the revised manuscript will better suit the Polymers but are happy to consider further revisions, and we thank you for your continued interest in our research.

We look forward to hearing from you in due time regarding our submission and to responding to any further questions and comments you may have.

Sincerely,

Corina Birleanu – first author

Marius Pustan and Grigore Pop – corresponding authors

31.08.2022

Round 2

Reviewer 1 Report

The discussion section needs to be modified by comparing the results of this study with the literature.

Author Response

Manuscript Polymers - Submission Polymers-1866217

Dear Allen Tian

Special Issue Editor

Thank you for giving me the opportunity to submit a revised draft of our manuscript entitled "Experimental investigation of the tribological behaviors of carbon fiber reinforced polymer composites under boundary lubrication" to Polymers.

We appreciate the time and effort that you and the reviewers have dedicated to providing your valuable feedback on our manuscript. We are grateful to the reviewers for their insightful comments on our paper. We have been able to incorporate changes to reflect most of the suggestions provided by the reviewers. We have highlighted the changes within the manuscript.

Here is a point-by-point response to the reviewers’ comments and concerns.

Comments from Reviewer 1

Reviewer 1

The discussion section needs to be modified by comparing the results of this study with the literature.

Response: Thank you for pointing this out. We have tried to respond to your requests, and we have improved the discussion section introducing the comparison of our results with those existing in the literature.

Finally, we can say that we have tried to incorporate all your suggestions throughout the manuscript. All the changes were made using Track Changes.

We hope the revised manuscript will better suit the Polymers but are happy to consider further revisions, and we thank you for your continued interest in our research.

We look forward to hearing from you in due time regarding our submission and to responding to any further questions and comments you may have.

Sincerely,

Corina Birleanu – first author

Marius Pustan and Grigore Pop – corresponding authors

02.09.2022

Reviewer 2 Report

Thanks for the authors' quick response.  Please double check minor errors and typos in the context.

Author Response

Manuscript Polymers - Submission Polymers-1866217

Dear Allen Tian

Special Issue Editor

Thank you for giving me the opportunity to submit a revised draft of our manuscript entitled "Experimental investigation of the tribological behaviors of carbon fiber reinforced polymer composites under boundary lubrication" to Polymers.

We appreciate the time and effort that you and the reviewers have dedicated to providing your valuable feedback on our manuscript. We are grateful to the reviewers for their insightful comments on our paper. We have been able to incorporate changes to reflect most of the suggestions provided by the reviewers. We have highlighted the changes within the manuscript.

Here is a point-by-point response to the reviewers’ comments and concerns.

Comments from Reviewer 2

Reviewer 2

Thanks for the authors' quick response.  Please double check minor errors and typos in the context.

General Response:

Thank you for the observation and we tried to improve all the errors and typos along the manuscript.

Finally, we can say that we have tried to incorporate all your suggestions throughout the manuscript. All the changes were made using Track Changes.

We hope the revised manuscript will better suit the Polymers but are happy to consider further revisions, and we thank you for your continued interest in our research.

We look forward to hearing from you in due time regarding our submission and to responding to any further questions and comments you may have.

Sincerely,

Corina Birleanu – first author

Marius Pustan and Grigore Pop – corresponding authors

02.09.2022
